# RestoreAgent: Autonomous Image Restoration Agent via Multimodal Large Language Models

**Haoyu Chen**[1], **Wenbo Li**[2], **Jinjin Gu**[3], **Jingjing Ren**[1], **Sixiang Chen**[1],
**Tian Ye**[1], **Renjing Pei**[2], **Kaiwen Zhou**[2], **Fenglong Song**[2], **Lei Zhu**[1,4*]

[1]The Hong Kong University of Science and Technology (Guangzhou)   [2]Huawei Noah's Ark Lab
[3]The University of Sydney   [4]The Hong Kong University of Science and Technology
Project page: https://haoyuchen.com/RestoreAgent

## Abstract

Natural images captured by mobile devices often suffer from multiple types of degradation, such as noise, blur, and low light. Traditional image restoration methods require manual selection of specific tasks, algorithms, and execution sequences, which is time-consuming and may yield suboptimal results. All-in-one models, though capable of handling multiple tasks, typically support only a limited range and often produce overly smooth, low-fidelity outcomes due to their broad data distribution fitting. To address these challenges, we first define a new pipeline for restoring images with multiple degradations, and then introduce RestoreAgent, an intelligent image restoration system leveraging multimodal large language models. RestoreAgent autonomously assesses the type and extent of degradation in input images and performs restoration through (1) determining the appropriate restoration tasks, (2) optimizing the task sequence, (3) selecting the most suitable models, and (4) executing the restoration. Experimental results demonstrate the superior performance of RestoreAgent in handling complex degradation, surpassing human experts. Furthermore, the system's modular design facilitates the fast integration of new tasks and models.

## 1  Introduction

Image restoration, a classical research area in computer vision, focuses on recovering high-quality images from degraded observations. Traditional methods are usually tailored to specific tasks like denoising [55, 61, 49, 29, 30, 12, 3], super-resolution [56, 34, 53, 4, 45, 47, 48], and deblurring [28, 22, 51, 32, 44, 17]. However, real-world images often suffer multiple simultaneous degradations. For example, a low-quality image may exhibit noise, blur, and rain concurrently. There may exist complex interactions and dependencies among different degradation phenomena, and each degradation may require distinct handling methods. The combination and sequence of these methods are crucial for the final restoration outcome. Recent advancements in the field have been driven by leveraging expert knowledge and developing all-in-one models. To provide a thorough understanding of this field and clarify our motivation, we present a detailed analysis below.

### 1.1  All-in-One Models

All-in-one models [38, 31, 24, 40, 33, 14, 27, 37, 1, 25] seek to use a single framework to handle multiple degradations simultaneously. By training on multi-task datasets, these models learn to manage various restoration tasks. However, several limitations continue to impede the practicality of these models in complex real-world scenarios:

**Restricted task scope.** All-in-one models often struggle to process degradations outside of their training data. Even for the same type of degradation, as shown in Figure 2 **a1**, these models may have difficulty effectively processing data if the degradation distribution varies between the training and testing sets. Given that existing models only cover a limited number of tasks, employing specialized single-task restoration models is often more flexible and effective.

---

*Lei Zhu (leizhu@ust.hk) is the corresponding author.

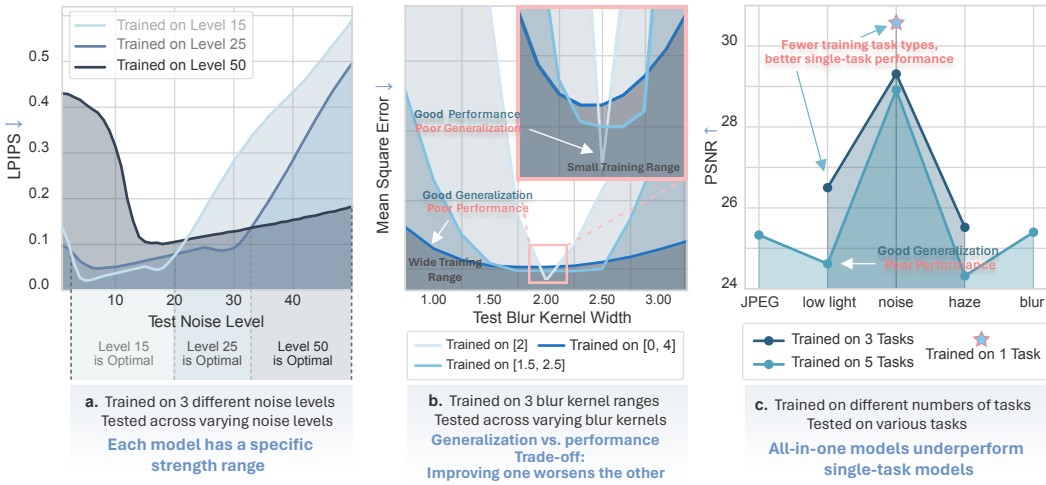

Figure 1: Limitations of all-in-one models. **(a)** Models trained on different noise levels excel in specific areas, so choosing models on demand leads to better results. **(b)** Models trained on a wider range of blur degradations offer improved generalization but compromised performance, showing a trade-off. **(c)** Multi-task models underperform on individual tasks compared to single-task models, illustrating that all-in-one models trade performance for generalization.

**Compromised performance.** All-in-one models often face trade-offs between generalization and restoration accuracy, as shown in Figure 1. While these models offer improved generalization across a broader range of degradation levels, their performance at specific levels may be compromised. Additionally, because they must handle multiple tasks with largely disparate degradation patterns, the performance for individual tasks may fall short, resulting in overly smoothed outputs. As illustrated in Figure 2 **a2**, single-task models typically outperform all-in-one models in most scenarios.

All-in-one models can, in fact, be integrated into an agent system comprising multiple models, thereby going beyond a single solution. Often, using task-specific models customized for particular degradations and then integrating them with an all-in-one model yields improved performance, as shown by the two examples in Figure 2 **a3**. This hybrid approach maintains the adaptability of all-in-one models while leveraging the strengths of specialized models.

## 1.2 Task-Specific Models

An alternative approach to using all-in-one models, which struggle to effectively address various types of degradation, is to combine several specialized task-specific models, each focusing on a specific degradation type. This modular strategy allows for a more targeted and efficient handling of the different degradations present in the input images. Superior results can be achieved because these specialized models excel in their respective areas.

### 1.2.1 Fixed or Random Execution Order

Current methods [50, 24, 14] typically detect the types of degradation in an image and apply the appropriate restoration models in a predetermined order, or manually selected by experts, or chosen at random. Nevertheless, there is a significant drawback to this approach: the processing order has a major impact on the final performance. A predetermined order, even if established by human experts, is not ideal and might fail to successfully restore the image, as demonstrated in Figure 2 **b**. Two primary causes can be identified for this.

**First, applying one restoration method can alter other degradation patterns**, rendering the following restoration models ineffective. For example, in an image with haze and rain, if haze is performed first (Figure 2 **b**), the dehazing model may address the blur but alter the rain distribution, thereby reducing the effectiveness of the deraining model.

**Second, removing some degradations can be challenging if other degradations have not been addressed first**. A common example is the enhancement of low-light images, which often requires denoising as a pre-processing step. Without prior denoising, the results of low-light enhancement are likely to be subpar. In Figure 2 **b**, we can observe that without prior denoising and deraining, the performance of the dehazing model is significantly compromised.

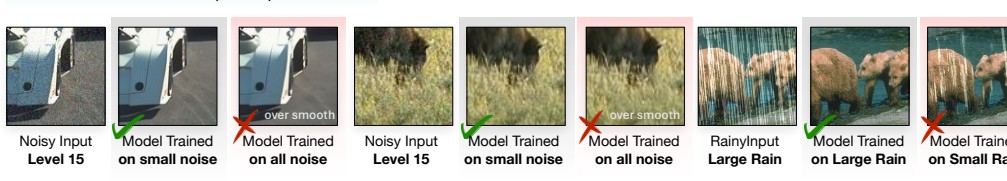

Figure 2: Limitation illustration of all-in-one models, fixed task execution order, and fixed model. Images with a pink background indicate negative examples.

In light of these findings, accurate identification of degradation patterns or careful testing of various task execution sequences is necessary for high-quality restoration. However, the search space grows significantly with the number of tasks. For example, there are 24 possible execution orders for 4 degradation types. Moreover, the number of permutations increases drastically when multiple models are available for a given task, leading to a significant rise in computational complexity.

### 1.2.2 Fixed or Random Model for a Single Task

In some scenarios, the system may opt to use a single model for a specific task or randomly select a model from a pool of available options [50]. However, this approach has significant drawbacks. Image restoration is a rapidly evolving field with various models tailored for a specific task, each with unique capabilities and areas of expertise for managing specific scenarios. Using a fixed model

or randomly selecting from a pool of models to process complex degradations can lead to suboptimal results. As illustrated in Figure 2 **c** and Figure 1 **a**, different denoising models excel at different noise levels. Choosing the right model is crucial for achieving the best result.

Manually selecting the best model is impractical due to the numerous combinations of task execution orders and available models. For example, with 3 degradation types and 3 models per type, there are 162 possible combinations. Evaluating these permutations is time-consuming and labor-intensive. Consequently, we often rely on one or two experienced-based solutions, which may not achieve the desired restoration effect.

### 1.3  RestoreAgent

In response to the aforementioned challenges, we propose RestoreAgent, an autonomous and intelligent image restoration system based on a multimodal large language model (MLLM). The MLLM's exposure to vast and diverse data endows it with superior generalization capabilities and has showcased remarkable performance in visual understanding and logical reasoning [46, 35, 18, 39, 43, 6, 62]. Furthermore, its flexibility facilitates the quick addition of new tasks, the definition of desired output formats, and easier human interaction.

Our framework offers the following functionalities:

**(1) Degradation Type Identification.** RestoreAgent automatically identifies the types of degradation present in an input image and determines the corresponding restoration tasks required.

**(2) Adaptive Restoration Sequence.** RestoreAgent goes beyond the constraints of predefined, human-specified model execution orders by dynamically evaluating the individual properties of each input image to decide the best sequence for utilizing the restoration models, thereby enhancing the overall efficiency of the image restoration procedure.

**(3) Optimal Model Selection.** Based on the specific degradation patterns in the input image, RestoreAgent dynamically selects the most appropriate model from the available pool for each restoration task, ensuring optimal performance.

**(4) Automated Execution.** Once the restoration sequence and model selection are determined, RestoreAgent autonomously executes the entire restoration pipeline without the need for manual intervention.

To this end, we start by defining the multi-degradation task and constructing a training dataset. This dataset includes paired degraded images (with one or more degradation types) and their ground truth (only for evaluation), along with the optimal task execution sequence and best model choice based on user-preferred goals. We then fine-tune MLLM to enable RestoreAgent to autonomously make task decisions and determine the optimal processing sequence and models. Experiments show that RestoreAgent's decision-making capabilities significantly outperform existing methods and human experts, achieving superior performance in recovering multi-degradation images. Notably, our method can quickly adapt to unseen tasks and models.

## 2  Related Work

### 2.1  Single-Task Image Restoration

In the field of single-task image restoration, numerous methods have focused on addressing specific types of image degradation. In denoising, models like DnCNN [59] and RNAN [63] have demonstrated significant effectiveness, among others. In deblurring, algorithms like DeblurGAN [28] and MIMO-UNet [13] and others stand out. For reducing JPEG artifacts, methods such as DCSC [19] and FBCNN [23] are particularly well-suited. Additionally, there are specialized methods for restoration under adverse weather conditions, including dehazing [52, 41], deraining [11, 7], and desnowing [8, 9, 5]. Each task often requires a specialized approach, leading to highly optimized algorithms that achieve sota performance for their specific targets compared to universal approaches.

### 2.2  All-in-One Image Restoration

Recent research has explored the development of All-in-One models that attempt to handle multiple degradation types simultaneously within a single framework. This kind of methods are trained to recognize and correct various forms of degradation concurrently. AirNet [31] featuring the contrastive-based degraded encoder and degradation-guided all-in-one restoration network. ADMS [38] uses adaptive filters to efficiently restore images with unknown degradations. TAPE [36] embeds a task-agnostic prior into a transformer, utilizing a two-stage process of pre-training and fine-tuning to enhance image restoration. PromptIR [40] and PIP [33] both utilize uniquely designed prompts to guide their networks. MiOIR [27] employs sequential and prompt learning strategies, which guide the

network to incrementally learn individual IR tasks in a sequential manner. MPerceiver [1] employs a multimodal prompt learning approach, utilizing Stable Diffusion priors to achieve high-fidelity all-in-one image restoration.

### 2.3 Agent in Image Restoration

Another research direction focuses on more intelligent image restoration systems. One class of such methods employs a toolbox approach to address image degradation separately. RL-Restore [57] prepares a toolbox consisting of small-scale convolutional networks, each specialized in different tasks. The system then learns a policy to select appropriate tools from the toolbox to progressively restore the quality of a corrupted image. However, RL-Restore supports only three types of degradation: blur, noise, and JPEG compression, which constrains its application scenarios and prevents it from utilizing new state-of-the-art models. Clarity ChatGPT [50] combines the conversational intelligence of ChatGPT with multiple image restoration methods. It automatically detects types of image degradation and selects appropriate methods to restore images. Conversely, Clarity ChatGPT identifies the presence of degradation but lacks research and design on the execution order of tasks and the optimal model selection for specific degradations in the input image.

Another class involves all-in-one approaches with degradation-aware guidance. InstructIR [14] pioneers a novel approach by utilizing human-written instructions to guide the recovery from various types of degradation. AutoDIR [24] automatically detect and restore images with multiple unknown degradations. LLMRA [25] generates text descriptions and encodes them as context embeddings with degradation information, and integrates these context embeddings into the restoration network. DA-CLIP [37] presents a degradation-aware vision-language model that guides the model to learn high-fidelity image reconstruction. For these all-in-one restoration assistant methods, inherent limitations exist in the practical applications of all-in-one models.

How to overcome these limitations, fully leverage the wide array of state-of-the-art models for different tasks available on the market, and determine the optimal execution sequence of image restoration tasks and the most suitable model for specific degradation pattern remain unexplored. This gap presents a significant opportunity for future research in intelligent image restoration systems.

## 3 RestoreAgent

In this section, we introduce RestoreAgent, an advanced image restoration agent designed to find the optimal model and execution sequence from a model pool to process images containing multiple degradations. RestoreAgent is built upon a state-of-the-art multimodal large language model, which possesses remarkable reasoning, generalization, and cross-modal understanding capabilities. By leveraging the model's ability to draw insights from vast amounts of multimodal data, establish connections between visual and textual information, and apply that knowledge to new contexts, RestoreAgent can effectively analyze complex image degradation scenarios, infer the most suitable restoration techniques, and generate optimal pipelines that combine the strengths of various specialized models. As a result, RestoreAgent consistently produces high-quality results.

In Section 3.1, we first define the problem of identifying the most effective combination and order of models from a given pool to restore images affected by various types of degradation. Next, in Section 3.2.2, we describe the process of constructing the training data for the RestoreAgent. The training data consists of paired samples, each containing a degraded image and its corresponding optimal restoration pipeline. Finally, we detail the training process of RestoreAgent, which involves fine-tuning the Llava-Llama3-8b model using the constructed training data in Section 3.2. By learning from these examples, RestoreAgent acquires the ability to analyze degraded images and generate optimal restoration pipelines based on the available model pool.

### 3.1 Problem Definition

We consider a comprehensive set of degradation types, denoted as $\mathcal{D} = \{d_1, d_2, \ldots, d_n\}$, where each $d_i$ represents a specific type of image degradation such as noise, JPEG artifacts, blur, rain streaks, fog, and low light conditions. For each degradation type $d_i$, we tailor a model library $\mathcal{M}_{d_i}$, comprising models $\{M_{d_i}^1, M_{d_i}^2, \ldots\}$. Each model $M_{d_i}^j$ is specifically trained to mitigate the effects of degradation $d_i$. The problem is formally defined as follows:

**Input:** A degraded image $I$ subjected to various degradation types $\mathcal{D}$. A model library $\{\mathcal{M}_{d_1}, \mathcal{M}_{d_2}, \ldots, \mathcal{M}_{d_n}\}$ tailored for processing $\mathcal{D}$. The user-provided scoring function $S$ for evaluating the image restoration process.

**Data Construction**

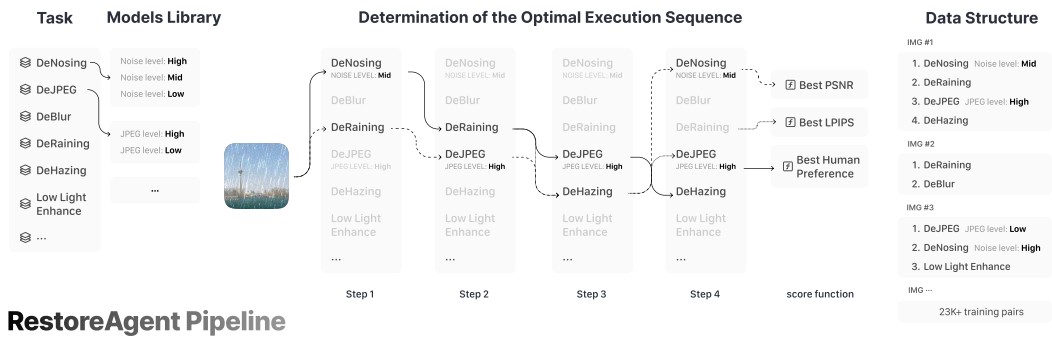

**RestoreAgent Pipeline**

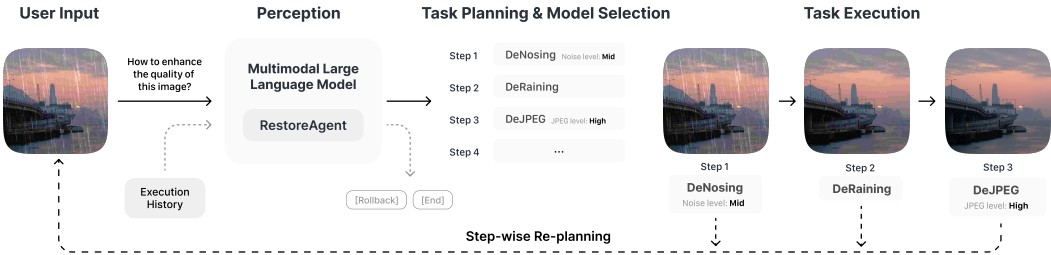

Figure 3: Illustration of the data construction workflow and RestoreAgent pipeline.

**Objective:** Identify the optimal model execution sequence $\sigma = (M_{a_1}^{b_1}, M_{a_2}^{b_2}, \ldots, M_{a_m}^{b_m})$ that maximizes the restoration quality $S$ of the degraded image $I$, where $a_i$ denotes the degradation type and $b_i$ represents the corresponding model. It is formulated as:

$$\sigma^* = \arg \max_{\sigma \in \mathfrak{S}(\mathcal{D}, \mathcal{M})} S(I, \sigma),$$

where $\mathfrak{S}(\mathcal{D}, \mathcal{M})$ represents the set of all possible sequences of degradation and model pairs.

By tackling this problem, we strive to identify the optimal combination of restoration sequence and model selections, ultimately enhancing the quality of images affected by multiple degradations in real-world settings, and thus providing a more effective and efficient solution for complex image restoration tasks.

## 3.2 RestoreAgent: An Advanced Image Restoration System

### 3.2.1 RestoreAgent Pipeline

We introduce an advanced image restoration agent, dubbed RestoreAgent, implemented using the state-of-the-art multimodal model Llava-Llama3-8b [46]. LoRA [21] is utilized to fine-tune both the vision and language modules. As shown in Figure 3, given a degraded input image, RestoreAgent can provide the best decisions, including which image restoration tasks need to be performed, the order of their execution, and which model is most suitable for each task. The model's input consists of a degraded image and the prompt such as `User: How to enhance the quality of this image? [Execution history: ...]`. In response, RestoreAgent generates an output sequence representing the optimal restoration pipeline, comprising a series of tasks, each associated with a specific model best suited to address particular degradation patterns. In our implementation, the output template is defined as: `Agent:1.<task name><model name>. 2.<task name><model name>. 3. ...`, ensuring interpretability and actionability.

RestoreAgent also supports an iterative step-wise decision-making process, reevaluating the state of the image after each restoration step. During this reassessment, the execution history is provided, offering valuable context for decision-making. This allows for real-time strategy adjustments based on cumulative effects and past actions. The system also features a rollback capability, enabling it to revert to a previous state if undesirable results are detected. This combination of iterative evaluation with historical context and rollback allows for finer control over the restoration process, facilitating mid-course corrections.

Table 1: Comparison of RestoreAgent with other decision-making strategies for multi-degraded image restoration. The "balanced" column represents the sum of the four normalized metrics, which is our score function to train the model. The "ranking" column indicates the ranking of the given decision among all possible decisions, with the total number of decisions for each test set provided. The final group presents the Average Result Across All Datasets, providing an overall performance.

| | Noise + JPEG | | | | | | Low Light + Noise | | | | | |
|---|---|---|---|---|---|---|---|---|---|---|---|---|
| | PSNR ↑ | SSIM ↑ | LPIPS ↓ | DISTS ↓ | balanced ↑ | ranking /17 | PSNR ↑ | SSIM ↑ | LPIPS ↓ | DISTS ↓ | balanced ↑ | ranking /10 |
| Random Order & Model | 24.52 | 0.7273 | 0.2889 | 0.2212 | 1.47 | 6.7 | 15.57 | 0.6541 | 0.4351 | 0.2588 | 1.98 | 3.9 |
| Random Oder + Predict Model | 25.24 | 0.7765 | 0.2327 | 0.1960 | 3.07 | 4.2 | 15.62 | 0.6887 | 0.3651 | 0.2283 | 3.03 | 3.0 |
| Random Model + Predict Order | 24.90 | 0.7568 | 0.2597 | 0.2132 | 2.03 | 6.0 | 17.57 | 0.7044 | 0.3685 | 0.2324 | 3.75 | 2.3 |
| Pre-defined Oder and Model | 25.29 | 0.7828 | 0.2366 | 0.2037 | 2.47 | 5.3 | 17.75 | 0.7098 | 0.3385 | 0.2260 | 3.93 | 2.1 |
| Human Expert | 25.06 | 0.7588 | 0.2551 | 0.2121 | 2.25 | 5.5 | 18.05 | 0.7239 | 0.3278 | 0.2220 | 4.29 | 1.9 |
| **RestoreAgent** | 25.32 | 0.7806 | 0.2308 | 0.1958 | 3.17 | 3.9 ↑1.6 | 17.80 | 0.7226 | 0.3259 | 0.2138 | 4.39 | 1.7 ↑0.2 |

| | Motion Blur + Noise + JPEG | | | | | | Rain + Noise + JPEG | | | | | |
|---|---|---|---|---|---|---|---|---|---|---|---|---|
| | PSNR ↑ | SSIM ↑ | LPIPS ↓ | DISTS ↓ | balanced ↑ | ranking /64 | PSNR ↑ | SSIM ↑ | LPIPS ↓ | DISTS ↓ | balanced ↑ | ranking /64 |
| Random Order & Model | 24.81 | 0.7816 | 0.2381 | 0.1747 | 2.32 | 19.5 | 25.64 | 0.7970 | 0.2412 | 0.2020 | 2.90 | 16.1 |
| Random Oder + Predict Model | 24.73 | 0.7787 | 0.2261 | 0.1684 | 2.69 | 16.1 | 25.67 | 0.8008 | 0.2368 | 0.1956 | 3.11 | 15.0 |
| Random Model + Predict Order | 24.95 | 0.7912 | 0.2263 | 0.1647 | 3.18 | 13.6 | 26.14 | 0.8074 | 0.2314 | 0.1996 | 3.49 | 13.3 |
| Pre-defined Oder and Model | 24.84 | 0.7895 | 0.2305 | 0.1662 | 2.97 | 15.0 | 25.80 | 0.7981 | 0.2360 | 0.2041 | 2.83 | 16.7 |
| Human Expert | 25.20 | 0.795 | 0.2205 | 0.1646 | 3.82 | 9.0 | 25.99 | 0.8063 | 0.2258 | 0.1992 | 3.58 | 12.6 |
| **RestoreAgent** | 25.16 | 0.7939 | 0.2042 | 0.1546 | 4.35 | 4.6 ↑4.4 | 26.38 | 0.8136 | 0.2200 | 0.1891 | 4.67 | 6.4 ↑6.2 |

| | Haze + Noise + JPEG | | | | | | Haze + Rain + Noise + JPEG | | | | | |
|---|---|---|---|---|---|---|---|---|---|---|---|---|
| | PSNR ↑ | SSIM ↑ | LPIPS ↓ | DISTS ↓ | balanced ↑ | ranking /64 | PSNR ↑ | SSIM ↑ | LPIPS ↓ | DISTS ↓ | balanced ↑ | ranking /287 |
| Random Order & Model | 18.98 | 0.7156 | 0.3267 | 0.2212 | 1.52 | 23.4 | 15.13 | 0.6300 | 0.4464 | 0.2800 | 1.28 | 102.5 |
| Random Oder + Predict Model | 19.00 | 0.7235 | 0.3133 | 0.2081 | 2.03 | 20.4 | 17.45 | 0.6897 | 0.3692 | 0.2400 | 2.86 | 72.8 |
| Random Model + Predict Order | 19.67 | 0.7653 | 0.2778 | 0.2010 | 2.95 | 15.9 | 19.79 | 0.7833 | 0.2815 | 0.1991 | 5.66 | 16.9 |
| Pre-defined Oder and Model | 19.47 | 0.7803 | 0.2641 | 0.1912 | 3.51 | 12.4 | 19.29 | 0.7785 | 0.2815 | 0.1974 | 5.502 | 26.1 |
| Human Expert | 19.50 | 0.7753 | 0.2703 | 0.1982 | 3.36 | 12.7 | 19.39 | 0.7802 | 0.2928 | 0.2043 | 5.503 | 21.3 |
| **RestoreAgent** | 19.55 | 0.7794 | 0.25663 | 0.1863 | 3.93 | 8.4 ↑4.3 | 19.72 | 0.7816 | 0.2741 | 0.1903 | 5.86 | 9.7 ↑11.6 |

| | Motion Blur + Rain + Noise + JPEG | | | | | | Average Result Across All Datasets | | | | | |
|---|---|---|---|---|---|---|---|---|---|---|---|---|
| | PSNR ↑ | SSIM ↑ | LPIPS ↓ | DISTS ↓ | balanced ↑ | ranking /287 | PSNR ↑ | SSIM ↑ | LPIPS ↓ | DISTS ↓ | balanced ↑ | ranking (%) |
| Random Order & Model | 21.96 | 0.6672 | 0.3366 | 0.2239 | 2.57 | 85.3 | 21.31 | 0.7139 | 0.3246 | 0.2241 | 1.92 | 34.7 |
| Random Oder + Predict Model | 22.11 | 0.6667 | 0.3038 | 0.2122 | 3.66 | 58.8 | 21.74 | 0.7385 | 0.2848 | 0.2045 | 2.89 | 26.1 |
| Random Model + Predict Order | 22.74 | 0.6996 | 0.2794 | 0.1979 | 5.39 | 24.7 | 22.42 | 0.7574 | 0.2750 | 0.2027 | 3.44 | 22.7 |
| Pre-defined Oder and Model | 22.35 | 0.6862 | 0.2858 | 0.1997 | 4.65 | 35.7 | 22.38 | 0.7639 | 0.2644 | 0.1986 | 3.48 | 22.1 |
| Human Expert | 22.96 | 0.7092 | 0.2861 | 0.2031 | 5.42 | 21.2 | 22.51 | 0.7634 | 0.2670 | 0.2014 | 3.73 | 19.5 |
| **RestoreAgent** | 22.95 | 0.7097 | 0.2615 | 0.1887 | 6.35 | 5.7 ↑15.5 | 22.61 | 0.7700 | 0.2513 | 0.1890 | 4.38 | 12.9 ↑6.6 |

### 3.2.2 Data Construction

To fully leverage the potential of multimodal large models, we construct a substantial dataset of paired training samples. The process begins with applying various types of degradation to an image. Subsequently, we determine the optimal restoration pipeline using model tools for processing. For each image undergoing multiple degradations, a comprehensive search is conducted to identify the best restoration pipeline, as shown in Figure 3. This involves generating all possible permutations of task execution sequences and model combinations, applying each pipeline to the degraded image, and assessing the quality of the restored outputs using a scoring function $S(I, \sigma)$. By comparing the scores of all permutations, the pipeline with the highest score is selected as the optimal processing strategy $\sigma$ for the given image. Users can choose from various image quality assessment methods as the scoring function, customizing the evaluation process to their specific needs. Additional details regarding dataset construction are provided in the supplementary materials.

## 4 Experiment

### 4.1 Experimental Settings

**Scoring function.** To construct a comprehensive evaluation system, we integrate multiple diverse metrics. Specifically, we first standardized each individual metric separately and then summed the standardized results. This process can be described as follows. Let $X_i$ represent the $i$-th metric. We standardize each metric by calculating its z-score: $Z_i = \frac{X_i - \mu_i}{\sigma_i}$ where $\mu_i$ is the mean and $\sigma_i$ is the standard deviation of the $i$-th metric. After standardizing all metrics, we aggregate the standardized scores to form the comprehensive evaluation score $S$: $S = \sum_{i=1}^{n} Z_i$ where $n$ is the total number of metrics. This method ensures that each metric contributes equally to the final evaluation, regardless of its original scale. Follow [26, 20], evaluation metrics primarily include PSNR, SSIM, LPIPS [60], and DISTS [16]. These metrics are widely recognized for their ability to comprehensively reflect the outcomes of image restoration. We also provided the results of models trained on individual metrics.

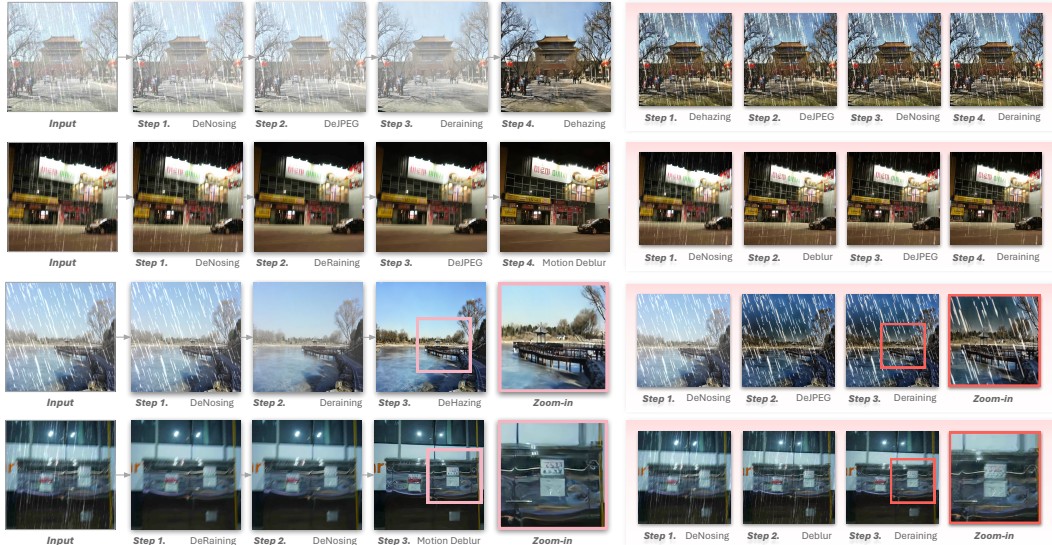

Figure 4: Illustrations of RestoreAgent's choices demonstrate that our approach predicts the correct task sequence. Images with a pink background show inappropriate decisions.

**Dataset and model tool settings.** To explore the feasibility of automating image restoration using multimodal models, we selected six distinct image restoration tasks: denoising, motion deblurring, deJPEG, deraining, dehazing, and low-light image enhancement. Each image in the dataset can exhibit up to four types of degradation. To validate the decision-making ability of the model when multiple models are available for a single task, we constructed three specialized models for the denoising task, and three models have different noise levels: low, medium and high noise. Similarly, for the deJPEG task, we developed models specifically designed to handle severe and mild JPEG compression artifacts. For the remaining tasks, each has a corresponding dedicated model. For the testing datasets, we assemble 200 images, mirroring the degradation types found in the training datasets, to facilitate evaluation. Detailed information is in the supplementary material.

## 4.2 Comparisons with Other Strategies

**Compared methods.** In this study, we conducted a comparative analysis of RestoreAgent against several alternative approaches:

- Random selection of both the task order and the models, assuming accurate determination of task types.
- Random task order, but models predicted by RestoreAgent.
- Random model selection, but task orders predicted by RestoreAgent.
- For all images, using the human expert's predefined order and models, assuming accurate task type determination.
- Human expert personally crafting a solution for each image, determining the task sequence and models for each task. This method represents the most common scenario in real-world applications, where a human decides how to restore an image on a case-by-case basis.

The human expert in this study has more than five years of research experience in low-level vision. Before crafting solutions, the expert familiarized themselves with each task degradation and the corresponding model's actual performance to ensure they could provide the best human-level solution.

**Results.** Table 1 reports the average metric results of our RestoreAgent and other decision-making approaches on seven different degradation combination datasets. As shown in Table 1, using a random order and model selection ranked lowest, achieving only a 34.7% performance rating among all possible strategies. By setting predefined sequences and models for image processing by human experts, traditional methods rank in the top 22.1% of all possible strategies. This demonstrates that experience-based predefined rules often used in practical applications are more effective than completely random strategies. Human experts making specific decisions for each test image can further improve upon predefined rules, increasing the ranking from 22.1% to 19.5%. This proves that using the same predefined rules to process all images is not optimal, while individualized decision-making for specific images can better enhance the effects. Then, the superior performance of our RestoreAgent (12.9%) over expert-based customization (19.5%) shows that automated and

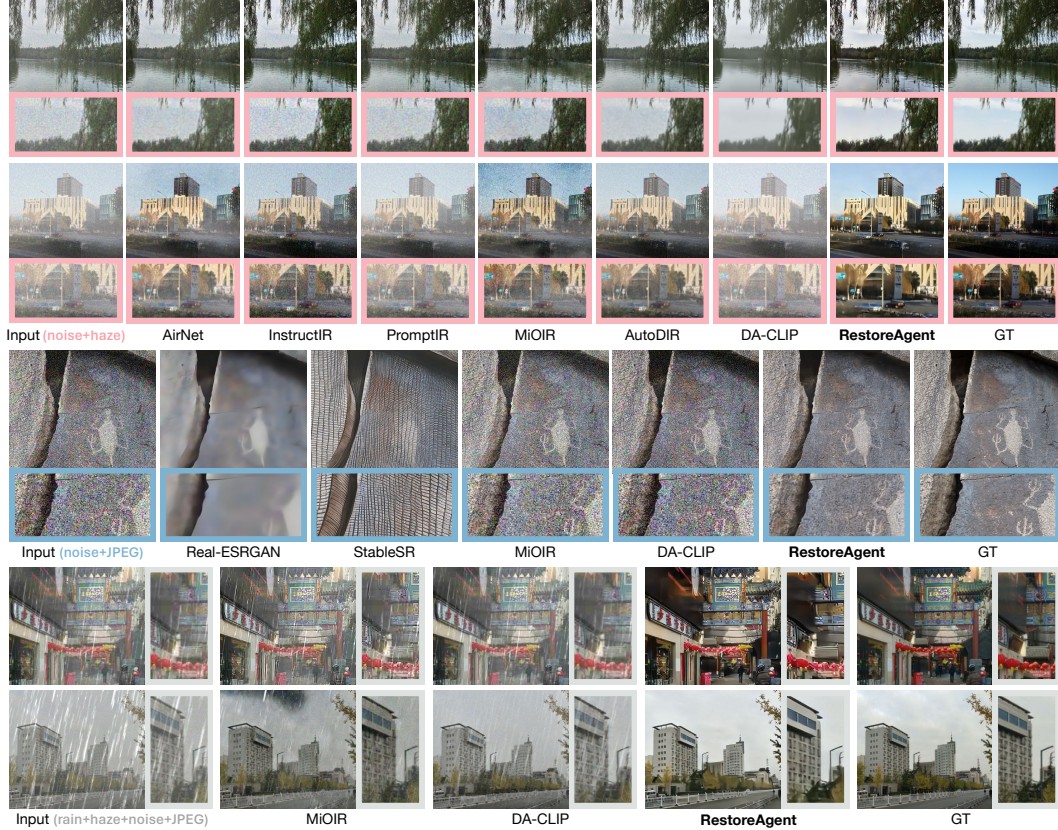

Figure 5: Visual comparisons with All-in-One Methods. To ensure a fair comparison, All-in-One methods are tested only on the degradation types and datasets they support. The all-in-one approach still lacks the ability to effectively handle images containing multiple types of degradation.

Table 2: Comparison of RestoreAgent with All-in-One methods for multi-degraded image restoration. We highlight best and second-best values for each metric.

| | noise + JPEG | | | | haze + noise | | | | rain + haze + noise | | | | rain + haze + noise + JPEG | | | |
|---|---|---|---|---|---|---|---|---|---|---|---|---|---|---|---|---|
| | PSNR ↑ | SSIM ↑ | LPIPS ↓ | DISTS ↓ | PSNR ↑ | SSIM ↑ | LPIPS ↓ | DISTS ↓ | PSNR ↑ | SSIM ↑ | LPIPS ↓ | DISTS ↓ | PSNR ↑ | SSIM ↑ | LPIPS ↓ | DISTS ↓ |
| Real-ESRGAN [48] | 23.43 | 0.7242 | 0.3022 | 0.2106 | - | - | - | - | - | - | - | - | - | - | - | - |
| StableSR [47] | 17.61 | 0.4464 | 0.3705 | 0.2124 | - | - | - | - | - | - | - | - | - | - | - | - |
| AirNet [31] | - | - | - | - | 17.56 | 0.5897 | 0.5569 | 0.2964 | 18.22 | 0.6767 | 0.4314 | 0.2336 | - | - | - | - |
| PromptIR [40] | - | - | - | - | 16.13 | 0.5428 | 0.6696 | 0.3544 | 17.81 | 0.7099 | 0.4506 | 0.2317 | - | - | - | - |
| MiOIR [27] | 23.98 | 0.6961 | 0.3266 | 0.2325 | 15.79 | 0.4790 | 0.7118 | 0.3628 | 16.22 | 0.6388 | 0.4719 | 0.2771 | 13.80 | 0.6410 | 0.4875 | 0.2939 |
| InstructIR [14] | - | - | - | - | 17.36 | 0.4288 | 0.7696 | 0.3646 | 19.45 | 0.6897 | 0.3994 | 0.2170 | - | - | - | - |
| DA-CLIP [37] | 22.47 | 0.6128 | 0.3525 | 0.2287 | 16.98 | 0.7061 | 0.3901 | 0.2737 | 15.44 | 0.6011 | 0.4597 | 0.2754 | 15.30 | 0.6863 | 0.3871 | 0.2627 |
| AutoDIR [24] | - | - | - | - | 17.51 | 0.6942 | 0.4248 | 0.2444 | 19.22 | 0.7705 | 0.3043 | 0.1802 | - | - | - | - |
| **RestoreAgent** | 25.32 | 0.7806 | 0.2308 | 0.1958 | 20.47 | 0.8053 | 0.2193 | 0.1758 | 19.53 | 0.8237 | 0.2166 | 0.1638 | 19.72 | 0.7816 | 0.2741 | 0.1903 |

data-driven decision-making in our method clearly outperforms traditional and experience-based human expert judgments. This is because human experts from their own experience can not make precise judgments about the advantageous scenarios of all models and the order of task execution, especially when numerous tasks and models are involved.

## 4.3 Comparisons with All-in-One Methods

To demonstrate the limitations of all-in-one methods in handling multi-degraded images, we compared our approach with various types of all-in-one models. To ensure a fair comparison, tests were only conducted on degradation types and datasets that these all-in-one models were trained to support. Moreover, we repeatedly run the all-in-one model as many times as the number of degradation types of the test images to fully leverage its capabilities, thus ensuring a fair comparison. The results are shown in Figure 5 and Table 2. Our RestoreAgent achieved a significant leading advantage across all tested degradation combinations. For the degradation types commonly encountered in traditional image super-resolution, such as noise and JPEG compression artifacts, our approach significantly outperformed established methods like Real-ESRGAN and the sota SR method, StableSR. For a broader range of degradation types, our method retained a considerable advantage. Among these all-in-one approaches, InstructIR and AutoDIR face two major issues: manually predetermined or

| | PSNR | | SSIM | | LPIPS | | DISTS | |
|---|---|---|---|---|---|---|---|---|
| | Value | ranking (%) | Value | ranking (%) | Value | ranking (%) | Value | ranking (%) |
| Pre-defined Oder and Model | 22.38 | 25.4 | 0.7639 | 26.5 | 0.2644 | 25.2 | 0.1986 | 22.1 |
| Human Expert | 22.51 | 20.5 | 0.7634 | 22.6 | 0.2670 | 25.1 | 0.2014 | 23.8 |
| RestorAgent - balanced | 22.61 | 19.9 | 0.7700 | 21.4 | 0.2513 | 17.1 | 0.1890 | 13.8 |
| RestorAgent - PSNR | 22.72 | 13.9 | - | - | - | - | - | - |
| RestorAgent - SSIM | - | - | 0.7763 | 16.5 | - | - | - | - |
| RestorAgent - LPIPS | - | - | - | - | 0.2477 | 13.4 | - | - |
| RestorAgent - DISTS | - | - | - | - | - | - | 0.1875 | 12.9 |

Table 3: RestoreAgent possesses the flexibility to adapt to various optimization objectives, enabling the generation of decision-making results tailored to specific target criteria.

Table 4: Analysis of the effect of training data size. Our model shows strong performance with smaller datasets (7k), but increasing the data volume (23k) results in further enhanced outcomes.

| | PSNR | SSIM | LPIPS | DISTS | balanced | ranking (%) |
|---|---|---|---|---|---|---|
| Random | 21.31 | 0.7139 | 0.3246 | 0.2241 | 1.92 | 34.7 |
| Human Expert | 22.51 | 0.7634 | 0.2670 | 0.2014 | 3.73 | 19.5 |
| **RestoreAgent** | | | | | | |
| - 7k | 22.63 | 0.7669 | 0.2568 | 0.1922 | 4.10 | 16.2 |
| - 14k | 22.57 | 0.7664 | 0.2528 | 0.1902 | 4.26 | 13.6 |
| - 23k | 22.61 | 0.7700 | 0.2513 | 0.1890 | 4.38 | 12.9 |

Table 5: Fast adaptation to new task (desnowing) in half an hour.

| | balanced | ranking /64 |
|---|---|---|
| Random | 0.54 | 27.1 |
| Pre-defined Order and Model | 3.82 | 9.2 |
| Human Expert | 3.91 | 8.5 |
| **RestoreAgent** | 4.23 | 4.3 |

Table 6: Step-wise planning.

| | balanced | ranking /64 |
|---|---|---|
| Human Expert | 5.42 | 21.2 |
| **RestoreAgent** | 6.35 | 5.7 |
| **RestoreAgent + Step-wise** | 6.38 | 4.5 |

randomly decided execution order, and using single model to address all types of degradations. These limitations often result in incomplete restoration, as depicted in Figure 5. These results underscore the limitations of all-in-one models, validating our initial hypothesis.

### 4.4  Adapting to Different Optimization Objectives

As discussed in the method, our proposed method can adapt to various optimization objectives, enabling the decision-making results tailored to specific target criteria. To verify it, we present the results of models trained with different individual metrics as the optimization objective in Table 3. The results indicate that when a model is trained with a single metric, the performance of the corresponding metric can be significantly improved compared to the balanced model. This showcases the adaptability and effectiveness of our method in catering to specific optimization goals.

### 4.5  Extending for New Tasks and Models

The proposed RestoreAgent demonstrates remarkable adaptability and extensibility, allowing for swift fine-tuning to accommodate new task types and incorporate additional models. To validate this capability, we introduced a new task, desnowing, along with its corresponding model. Building upon the RestoreAgent previously trained on six tasks, we performed rapid fine-tuning by integrating the desnowing task. Within thirty minutes, our model achieved exceptional performance on the new task type. As shown in Table 5, our approach quickly surpassed human expert-level proficiency on the new task and model. This validation underscores the practical value of our method, allowing efficient integration of additional tasks with minimal resource expenditure.

### 4.6  Step-wise Re-planning and Rollback

As mentioned in Section 3.2, RestoreAgent supports iterative decision-making with historical context awareness. It dynamically adjusts strategies during image restoration, reassessing image state after each step and rolling back if needed. As demonstrated in Table 6, we conducted experiments on a complex dataset incorporating four distinct types of image degradation: Motion Blur, Rain, Noise, and JPEG compression. Results show that while the single prediction approach performs well, iterative step-wise replanning further enhances restoration outcomes, allowing for precise control and mid-course corrections. The initial decision's performance is already strong, step-wise replanning thus offers incremental yet valuable improvements to an already effective process.

## 5  Conclusion

Our research identifies key factors in processing multi-degraded images, including task execution order, model selection, and the limitations of the all-in-one approach. Based on these insights, we present RestoreAgent, an agent model that makes intelligent processing decisions based on image degradation and user objectives. Experiments show that our pipeline outperforms the all-in-one method and surpasses human experts in decision-making performance.

**Acknowledgments.** This work is supported by the Guangzhou-HKUST(GZ) Joint Funding Program (No. 2023A03J0671), the Guangzhou Municipal Science and Technology Project (Grant No. 2024A04J4230), Guangdong Provincial Key Lab of Integrated Communication, Sensing and Computation for Ubiquitous Internet of Things(No.2023B1212010007), and the National Natural Science Foundation of China (Project No. 61902275).

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

# A    Appendix / supplemental material

## A.1    Model Tool Settings

As shown in Table 7, for the tasks of denoising and deJPEG, as well as deraining, we employ Restormer [58] as our model. For dehazing, we utilize RIDCP [54], while for motion deblurring, we use DeblurGANv2 [28]. For desnowing, we implement Snowformer [10]. For low-light enhancement, we use Retinexformer [2]. It is noteworthy that the models we are using are not the latest state-of-the-art models, indicating that there is significant room for improvement in our models.

A crucial consideration in image restoration is the limited generalization capability of many current models, which often fail to maintain performance when faced with subtle variations in image degradation. This necessitates the selection of more robust models. For example, in our approach to denoising, we enhance model generalization by incorporating not only Gaussian noise but also random blur and other noise types during training. This strategy enables the model to address more complex degradation scenarios effectively.

Table 7: Model tools for different restoration tasks.

| Task | Model Tools |
|---|---|
| Gaussian denosing | Restormer (trained on large noise level) |
| | Restormer (trained on medium noise level) |
| | Restormer (trained on small noise level) |
| DeJPEG | Restormer (trained on high quality factor) |
| | Restormer (trained on low quality factor) |
| Dehazing | RIDCP [54] |
| Deraining | Restormer [58] |
| Motion deblurring | DeblurGANv2 [28] |
| Low-light enhancement | Retinexformer [2] |

Table 8: Testset details.

| Degradatio | Number of Images |
|---|---|
| Noise + JPEG | 50 |
| Noise + Low light | 30 |
| Motion Blur + Noise + JPEG | 30 |
| Rain + Noise + JPEG | 20 |
| Haze + Noise + JPEG | 30 |
| Haze + Rain + Noise + JPEG | 20 |
| Motion Blur + Rain + Noise + JPEG | 20 |
| **Total** | **200** |

## A.2    Dataset Construction Details

Figure 6: Five scenarios for dataset construction and their corresponding examples.

Figure 6 illustrates 5 scenarios incorporated into our dataset, designed to enhance the versatility and robustness of the RestoreAgent model:

(1) Once we obtain a degraded image along with its corresponding optimal decision results, we can construct the primary part of our dataset. This part consists of degraded images in their original, unprocessed state. For these inputs, the RestoreAgent receives a prompt: "How to enhance the quality of this image? Execution history: None." This scenario trains the model to formulate comprehensive enhancement strategies from scratch, encompassing multiple restoration steps. This part of the data exceeds 23k pairs.

(2) To foster dynamic decision-making capabilities, we introduce a second category of training instances. Here, the input comprises partially processed images (e.g., after denoising) along with their execution history. This approach enables the RestoreAgent to adapt its predictions based on intermediate results, promoting a more flexible and context-aware enhancement process.

(3) The third scenario addresses situations where the model identifies suboptimal outcomes from a particular enhancement step. In such cases, the RestoreAgent is trained to output "Rollback," indicating the need to revert to a previous state and recalibrate its strategy. This feature is crucial for maintaining high-quality outputs and avoiding the propagation of errors through the enhancement pipeline. We select from erroneous paths (the decisions with the worst metric results) to construct this portion of the paired data, as the worst paths require a rollback.

(4) Following a rollback event, our fourth data category provides the model with information about the specific step that triggered the rollback. This guidance is essential in preventing the model from repeating ineffective procedures, thus streamlining the enhancement process and improving efficiency.

(5) The final scenario in our training regime represents fully processed images that require no further enhancement. In these instances, the RestoreAgent is trained to recognize optimal image quality and output "Stop", effectively terminating the enhancement sequence.

By incorporating these diverse scenarios, we aim to develop a highly adaptive and efficient image restoration system capable of addressing a wide array of real-world image degradation challenges. For computational efficiency, unless specifically mentioned otherwise, our default experiments are based on a single planning for the initial image rather than using iterative step-wise replanning.

### A.3    Testset details

The specific details of our test set are presented in Table 8, which demonstrates our construction of various combinations of degradation types. Each image in the set contains a minimum of one and a maximum of four types of degradation, with the entire set comprising 200 images.

### A.4    Training Setups

In this study, we incorporate the CLIP pre-trained Vision Transformer (ViT-L/14) [42] as the image encoder to convert input images into visual tokens. For the language model, we utilize the Llama3-7B [46]. Despite their capabilities, pre-trained LLMs fail to provide accurate responses without dataset-specific fine-tuning. To address this, we adopt LoRA [21], a fine-tuning technique that efficiently modifies a limited number of parameters within the model. Following [21], we apply LoRA to adjust the projection layers in all self-attention modules of both the vision encoder and the LLM, thereby generating our RestoreAgent. We employ the Xtuner framework [15] to facilitate the training process. For our experimental setup, we configure the LoRA rank to 16. The RestoreAgent undergoes training across ten epochs on 4 NVIDIA RTX A100 GPUs, with a batch size of 32. We employ the Adam optimizer and a learning rate of 0.00002. The total duration of the training process approximates ten hours.

### A.5    Analysis

Figure 4 and 8 illustrate RestoreAgent's decision-making process and the importance of model selection, respectively. Figure 6 further demonstrates why human decision-making often yields suboptimal results in image restoration tasks. Figure 8a exemplifies the nuanced challenges in degradation assessment. Despite identical backgrounds and degradation types, subtle variations in degradation features lead to divergent optimal restoration sequences. For instance, the sequence

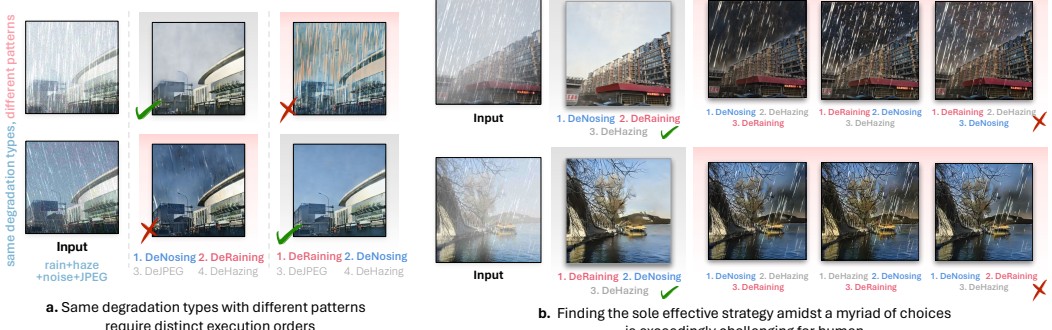

Figure 7: Challenges in human expert decision-making. This figure illustrates the difficulty faced by human experts in discerning minute differences between degradation patterns, leading to suboptimal restoration strategies.

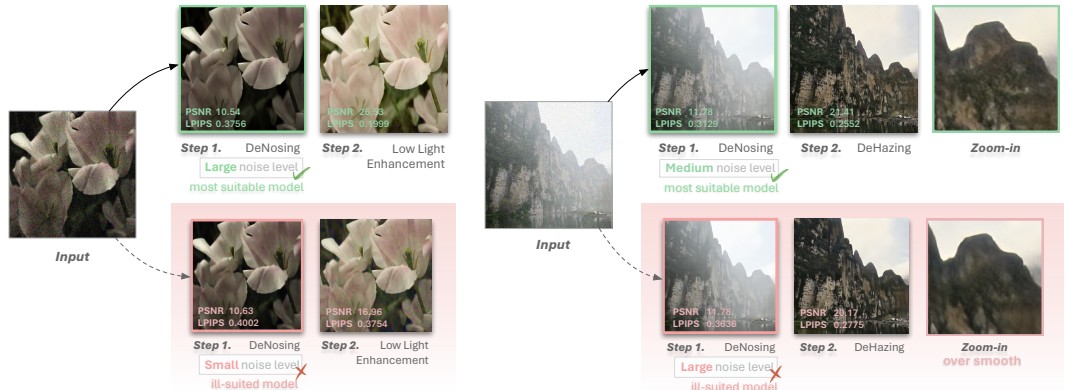

Figure 8: Examples of model decisions made by RestoreAgent. This figure demonstrates how choosing the appropriate model for a specific restoration task significantly affects the outcome quality. We present PSNR and LPIPS metrics for each image. Images with a pink background indicate examples of inappropriate decisions (zoom-in for better view).

"DeNoising → DeRaining → DeJPEG → DeHazing" proves effective for the upper row images but fails for the lower row. Conversely, the sequence "DeRaining → DeNoising → DeJPEG → DeHazing" yields optimal results for the lower row but is suboptimal for the upper row. This dichotomy underscores the difficulty human experts face in discerning minute degradation differences, thereby compromising effective decision-making.

The complexity of optimal restoration sequencing is further accentuated in Figure 8b. Here, we demonstrate scenarios where only one specific sequence among numerous permutations yields satisfactory results. This observation highlights the formidable challenge posed to human decision-makers in identifying the singular effective restoration pathway amidst a multitude of possibilities.

These findings collectively emphasize the superiority of automated, data-driven approaches in navigating the intricate landscape of image restoration. The RestoreAgent's ability to discern and adapt to subtle degradation variations surpasses human capabilities, particularly in scenarios where the optimal restoration sequence is non-intuitive and highly specific to individual image characteristics.

## A.6 Discussion

**Comparison with assistants with all-in-one models.**  Assistants that employ unified models, such as LLMRA [25] and AutoDIR [24], attempt to handle diverse tasks, degradation patterns, and intensities using a single model. As discussed in Section 1.1, these all-in-one models face significant challenges, including restricted task scope and compromised performance, which greatly limit their effectiveness in real-world applications. Conversely, our method leverages various model experts to address specific situations, the upper bound of our pipeline is determined by the latest SOTA models, allowing us to maximally leverage the latest advancements in the field without being constrained by

the limitations of an all-in-one model. Furthermore, as detailed in Section 4.4, our RestoreAgent exhibits high efficiency in incorporating new tasks and models, showcasing greater flexibility.

**Comparison with assistants with tool use.** Image restoration assistants that utilize tool libraries, such as Clarity ChatGPT [50] and RL-Restore [57]. Clarity ChatGPT merely identifies the degradation in images, follows a rigid execution strategy, lacking the ability to make dynamic decisions on task execution order and select the best model. As discussed in Section 1.2.1 and 1.2.2, an inappropriate task execution sequence and model selection can leading to lower performance in subsequent operations. RL-Restore, on the other hand, uses reinforcement learning for sequence decision-making and model selection. However, its task definition is overly simplistic, limited to three degradation types (noise, blur, and JPEG) with a narrow degradation range. Also, training reinforcement learning-based methods is more challenging and may result in lower precision, making it difficult to achieve high performance in complex and varied scenarios. Conversely, the integration of a comprehensive task definition with advanced multimodal models allows our method to effectively manage various degradation types and intensities. This adaptability enhances its efficacy, positioning our approach as a promising solution for image restoration tasks.

### A.7 Alation Study

**Training data amount.** To investigate the effect of training data volume on our method, we evaluated the performance of the RestoreAgent model trained on datasets consisting of 7,000, 14,000, and 23,000 data pairs; see Table 4 The results demonstrate that even with the smallest dataset of 7k pairs, our RestoreAgent achieves superior performance over both random and human expert benchmarks. More notably, the training data volume increasing from 7k to 14k incurs a substantial performance improvement with the ranking percentage decreasing from 16.2% to 13.6%. With 23k data pairs, the performance further improves, achieving a ranking percentage of 12.9%. This indicates that using more training data boosts our RestoreAgent model. These findings emphasize the robustness of our approach, demonstrating that while larger datasets do enhance performance, our model already provides significant benefits even with relatively smaller datasets.

### A.8 Limitation and Future Work

The primary limitation of our study is the confined scope of models and tasks examined. While our research offers valuable insights into RestoreAgent's performance across several degradation scenarios, it does not encompass the full spectrum of restoration models or image degradation tasks currently available.

Another limitation pertains to the limited generalization capability of current image restoration models. These models often exhibit a notable decrease in performance or fail to respond adequately when faced with even minor variations in image degradation patterns. This limitation greatly narrows our selection of model tools, requiring us to choose more robust and generalizable model tools. The challenge underscores a critical need in the field of image restoration: future models must go beyond simply overfitting training data. Rather, they should exhibit better generalization and increased efficiency in handling real-world degradation cases.

Our future work will focus on significantly expanding the range of image restoration models incorporated into our multimodal large language model. This expansion aims to enhance RestoreAgent's capabilities across a broader scope of restoration tasks and degradation types. By integrating a more diverse set of state-of-the-art models, we seek to create a more comprehensive and versatile restoration framework.

