# OpenReview forum: "RestoreAgent: Autonomous Image Restoration Agent via Multimodal Large Language Models"
_NeurIPS.cc/2024/Conference — NeurIPS 2024 poster_

### Official Review · Reviewer_4Dbc · 2024-06-19

**Soundness:** 3
**Presentation:** 4
**Contribution:** 3
**Rating:** 7
**Confidence:** 5

**Summary:**

For real-world images corrupted by multiple simultaneous degradations, this paper first analyzes the limitations of using all-in-one restoration models and various task-specific models. The authors then introduce RestoreAgent, which automatically identifies the types of degradation in a degraded image, determines the sequence of restoration tasks, and selects suitable models from the model pool. RestoreAgent presents an automated restoration pipeline that requires only an input image and a general human instruction, without any prior knowledge of the involved degradation tasks or manually predefined task sequences.

**Strengths:**

1. The paper comprehensively analyzes the challenges and limitations of employing all-in-one models and multiple task-specific expert models with fixed or random task sequences, as well as fixed or random models for each task.
2. The authors evaluate various configurations of RestoreAgent using diverse objective image quality metrics (PSNR, SSIM, LPIPS, DISTS, and their combinations), all of which outperform the human expert model on the corresponding metric.
3. RestoreAgent exhibits the scalability by extending to new tasks and models with minimal computational resource.
4. The presentation, including writing, analysis, and visualization, is clear and easy to follow.

**Weaknesses:**

1. Incomplete descriptions about data construction.

- Authors randomly select up to four types of degradation from a degradation set (noise, blur, JPEG, rain, haze, and low-light) to construct paired training data. According to data synthesis strategies in [1,2], JPEG compression is typically performed after noise and blur, and in the final order. Is the degradation order of JPEG compression in this paper the same? If not, the authors should discuss the reasonableness of random sampling.

- What are the components of 23k paired data? One degraded image for each high-quality image or many degraded versions for each high-quality image?

- What is the configuration in ablation studies about training data amount? Simultaneously scaling up low & high-quality images or synthesizing more low-quality images for each high-quality image? If it’s the former, will increasing the number of degraded images while keeping the number of high-quality images unchanged improve performance?

2. Inference time for input images with diverse resolution.
- The authors are suggested to report the running time for input images of various resolutions. This should include the total time, the running time for the RestoreAgent, and the running time for the subsequent restoration models. The reviewer is curious whether the agent's response time exceeds that of the restoration models when processing high-resolution images, such as those with 4K resolution.

3. Scalability for new tasks and models.
- Section 4.5 demonstrates that the proposed RestoreAgent can extend to new tasks and models in just half an hour, surpassing human expert-level performance on the new task. However, it is unclear whether adaptation to the new task results in performance degradation on prior tasks, similar to the catastrophic forgetting problem in continual learning. The authors are encouraged to report the performance of the fine-tuned model on the previous tasks to address this concern.

[1] Wang X, Xie L, Dong C, et al. Real-esrgan: Training real-world blind super-resolution with pure synthetic data[C]//Proceedings of the IEEE/CVF international conference on computer vision. 2021: 1905-1914.

[2] Zhang K, Liang J, Van Gool L, et al. Designing a practical degradation model for deep blind image super-resolution[C]//Proceedings of the IEEE/CVF International Conference on Computer Vision. 2021: 4791-4800.

**Questions:**

Addressing concerns in the Weaknesses with thorough explanations and additional experiments would significantly enhance my confidence in this work. A satisfactory response to these points may lead to a reconsideration of the current evaluation.

**Limitations:**

The manuscript includes the checklist guidelines.

---

> ### Author Rebuttal · Authors · 2024-08-04
>
> `Q1`: **Degradation order of JPEG compression.**
>
> Thank you for bringing up this important point regarding the degradation order. In our study, the order of JPEG compression is not fixed and is entirely random, unlike the sequence suggested in references [1,2]. The strategy of placing JPEG compression at the end, as mentioned by the reviewer, represents a subset of the data we constructed. Therefore, our dataset inherently includes this configuration while also supporting a broader range of degradation combinations.
>
> Our primary objective is to validate the robustness and feasibility of our proposed pipeline. By incorporating a wider variety of degradation sequences, we can more effectively demonstrate the generalizability and effectiveness of our method across different scenarios. Moreover, the specific degradation order can be flexibly defined and adjusted based on user requirements, ensuring adaptability to various applications.
>
>
> `Q2`: **What are the components of 23k paired data? One degraded image for each high-quality image or many degraded versions for each high-quality image?**
>
>
> Our dataset comprises one degraded image for each high-quality image. This means that each image pair consists of a unique degraded image and its corresponding high-quality counterpart, ensuring diverse image backgrounds. Specifically, we utilized images from the DIV2K and Flickr8K datasets, as well as the RESIDE dataset to construct images containing haze. Additionally, the LOLv2 Real dataset was employed to create low-light images.
>
>
>
> `Q3`: **What is the configuration in ablation studies about training data amount?**
>
>
> As previously mentioned, each high-quality image in our dataset corresponds to a single degraded image. Therefore, when we scale up or down the training data, both low-quality and high-quality images increase or decrease simultaneously.
>
> Increasing the number of degraded images while keeping the number of high-quality images unchanged would indeed improve performance. As discussed in our manuscript, image degradation is complex, and even minor variations can lead to different degradation characteristics, requiring varied handling strategies. Thus, adding more degraded images extends the range of degradations the model is exposed to, enhancing its ability to manage a broader spectrum of degradation scenarios.
>
>
>
> `Q4`: **running time**
>
> The RestoreAgent in our approach is constructed based on a multimodal large language model. Specifically, the computational time for the RestoreAgent is approximately 0.4 seconds. Therefore, the additional running time introduced by the RestoreAgent is minimal. In our experiments, the image patch size used is 512x512 pixels. In practice, the MLLM resizes all inputs to a fixed size, such as 224x224. For processing 4K resolution images, there are several strategies to consider:
> (1) Single Patch Representation: Often, image degradation across different regions is uniform. In such cases, a single patch can represent the overall degradation of a 4K image.
> (2) Patch-wise Processing: The image can be divided into smaller blocks, with each block represented by a patch.
> (3) Resizing: The 4K image can be resized appropriately before making decisions using the multimodal model.
> Given these strategies, decision-making for 4K images does not significantly increase the required time.
>
> Regarding the running time of the restoration models, it is important to note that our pipeline utilizes restoration models based on user preferences. Consequently, we can select models of varying sizes and running times according to specific needs. Therefore, concerns about the running time of the restoration models may not be necessary as the flexibility in our pipeline allows for optimization and customization based on user requirements.
>
>
>
>
> `Q5`: **it is unclear whether adaptation to the new task results in performance degradation on prior tasks, similar to the catastrophic forgetting problem in continual learning.**
>
>
> Thank you for pointing out this important concern. As shown in Table 1, we report the performance of the fine-tuned model on previous tasks after adaptation. The changes in performance are minimal. We believe that these minor variations in performance are primarily attributable to the **inherent randomness in the training process**. We believe that with careful tuning, the model's performance can be further enhanced.
>
> This demonstrates a crucial point: the current capabilities of multimodal large language models are more than sufficient to handle this task. Their robust reasoning abilities support the management of scenarios far more complex than those presented in our experiments.
>
> We believe one reason for this is that the newly added degradation task does not conflict with the existing tasks; in some aspects, it is very similar. For example, in the combination of snow + noise + JPEG, the original model already has extensive knowledge of handling noise + JPEG, which can be easily applied to the new combination.
>
> We are confident that in more complex scenarios (with more restoration tasks and more restoration models), RestoreAgent will demonstrate its significant advantages compared with human. In such complex scenarios, human decision-making will be less effective since the space is much bigger, whereas the powerful capabilities of learning-based multimodal large language models will exhibit even greater superiority.
>
>
>
>
>
>
> |       method   | Average Ranking on All Previous Datasets | Ranking on the New Task |
> |---|---|---|
> | Human Expert            | 19.5%    | 21.2 / 64 |
> | RestoreAgent             | 12.9%    | -    |
> | RestoreAgent + New Task  | 13.1%  | 4.5 / 64  |

---

> > ### Comment · Reviewer_4Dbc · 2024-08-12
> >
> > Thanks to the authors for their detailed responses. After considering the other reviews and the replies provided, I can confirm that the authors have addressed all my concerns about data construction, catastrophic forgetting, and running time. Additionally, the authors provided potential solutions for handling high-resolution inputs. Thus, I raised my final rating to "accept." I also encourage the authors to explore the incorporation of RLHF and human preference optimization in future updates. I believe that this agent-based paradigm has the potential to spark a new wave of research in the low-level vision community.

---

### Official Review · Reviewer_Ux72 · 2024-07-06

**Soundness:** 3
**Presentation:** 3
**Contribution:** 3
**Rating:** 5
**Confidence:** 4

**Summary:**

This paper proposes a new pipeline to address multiple degradation, like noise, blur and low light. Besides, a RestoreAgent with multimodal large language models is introduced to assess the type and extent of degradations in the input images and perform dynamic restorations.

**Strengths:**

1. The paper is well-written and well-organised.
2. The whole pipeline seems to be novel and reasonable.
3. The method achieves SOTA performance on several benchmarks and different degradation tasks.

**Weaknesses:**

The overall motivation of this paper is commendable, but I have a few concerns:

1. The author mentions that RestoreAgent can autonomously assess the type and extent of degradation in input images and perform restoration. This strategy is interesting. However, I am wondering how the order of different enhancement techniques is defined. For example, if the input has noise and rain streaks, how is the order of dehazing and denoising techniques determined? Will this affect performance?

2. In contrast to other image enhancement techniques, the proposed RestoreAgent should first find a suitable restoration task and then select the most appropriate model to enhance the quality of the input. Therefore, I am concerned whether this process will increase the inference time. The authors should provide some computational analysis.

3. The enhancement capabilities of this work rely heavily on existing enhancement frameworks. If existing frameworks cannot work well in some cases, such as extreme noise effects, I guess the proposed RestoreAgent may also fail. Is this true? If so, I suggest the authors mention this in the limitations section.

4. The explanation of "ranking" and "balanced" in Table 1 is still unclear. The authors should clarify the definitions of these terms.

5. It would be better to show more visual comparisons of the RestoreAgent.

**Questions:**

Please see weaknesses.

**Limitations:**

Limitations are not mentioned in the paper.

---

> ### Author Rebuttal · Authors · 2024-08-04
>
> `Q1`: **How the order of different enhancement techniques is defined. For example, if the input has noise and rain streaks, how is the order of dehazing and denoising techniques determined? Will this affect performance?**
>
> Thank you for highlighting this important aspect. The order of applying different enhancement techniques can significantly impact the restoration results, as illustrated in Figure 2 (b.1, b.2) of our manuscript.
>
> **Impact of Degradation Order on Patterns**
>
> When constructing training data, the sequence of applying degradations like noise, JPEG compression, and blur affects the resulting degradation patterns. For instance, adding noise before JPEG compression/blur alters the noise characteristics, making it smoother due to the compression/blur effects. Conversely, adding JPEG compression/blur before noise results in unaltered noise characteristics. This interplay between degradations creates distinct patterns based on their application order. The MLLM can identify different task sequences through degradation patterns.
>
> **Influence of Restoration Order on Performance**
>
> Similarly, the order of applying restoration techniques affects the degradation characteristics. For example, applying a deblurring model first can sharpen noise and rain streaks, altering their features. Conversely, applying a denoising model first can reduce noise but may impact rain streak features, making subsequent deraining less effective. An interesting case is when noise and rain streaks coexist: low noise levels may not impact deraining, but high noise levels can render deraining ineffective. Conversely, denoising first might alter rain streak features, affecting deraining performance.
>
> Our proposed RestoreAgent, being a learning-based method, is trained on extensive data to recognize degradation patterns and make informed decisions based on subtle variations. It autonomously determines the optimal sequence of restoration steps to effectively remove degradations. This is achieved through a comprehensive understanding of how different degradations and restoration techniques interact.
>
>
>
> `Q2`: **Inference time.**
>
> RestoreAgent is built on a MLLM. In our experiments. The image patch size we used is 512x512 pixels. In practice, the MLLM resizes all inputs to a fixed size, such as 224x224, and the inference time of RestoreAgent is approximately 0.4 seconds in RTX4090. The increase in inference time is very minimal.
>
>
> `Q3`: **If existing frameworks cannot work well in some cases, such as extreme noise effects, I guess the proposed RestoreAgent may also fail. Is this true? If so, I suggest the authors mention this in the limitations section.**
>
> We appreciate the reviewer's concern regarding the generalization capability of our model when faced with degradations outside the predefined scope. Here, we would like to further elaborate on this aspect.
>
> First of all, it is true that there might be generalization issues when encountering unseen types or levels of degradation. We acknowledge this limitation and have discussed it in detail in our manuscript.
>
> However, the primary and most significant contribution of our work is to demonstrate the feasibility and effectiveness of the proposed pipeline. The essence of our work lies in utilizing multimodal large models as agents for image restoration, which has shown promising results across various conditions. Our experiments show that the RestoreAgent performs exceptionally well in most complex scenarios, thus proving its viability.
>
> In addition, our approach allows for flexible integration of different tools. If the model encounters degradations beyond the predefined scope, additional tools can be seamlessly incorporated. This adaptability ensures that our pipeline remains robust even when faced with unexpected degradation types.
> Essentially, the generalization issue is more related to the specific image restoration models (tools) used within the pipeline, rather than the pipeline itself.
>
> Lastly, one of our future research directions is to significantly expand the scope of restoration tasks and incorporate a greater variety of restoration models. By doing so, we aim to enhance the system's capability to handle real-world complex degradations more effectively.
> This expansion will involve integrating models trained on a broader spectrum of degradation types and more generalizable restoration models, ensuring that the pipeline can generalize better to unseen conditions.
>
> `Q4`: **The explanation of "ranking" and "balanced" in Table 1 is still unclear. The authors should clarify the definitions of these terms.**
>
>
> We have added the following clarification to the paper:
>
>
> **The explanation of "ranking":**
>
> We've clarified the ranking system:
>
> For individual test sets:
> X/Y format: X is the average ranking, Y is total combinations.
> Example: 6.4/64 means average 6.4th best out of 64 options.
>
> For overall average:
> We use percentages due to varying combination counts of diffrent restoration tasks.
> Process: Calculate percentage ranking per set, then average.
> Example: 12.9% means top 12.9% on average across datasets.
>
> **The explanation of "balanced":**
>
> We have also provided a detailed explanation of "balanced" in Section 4.1 of our manuscript, specifically within the Scoring Function subsection. Balanced refers to the sum of the PSNR, SSIM, LPIPS, and DISTS metrics after each has been standardized. To ensure clarity, we have also added additional explanatory notes directly in Table 1. We hope these changes address your concerns.
>
> `Q5`: **Show more visual comparisons.**
>
> Thank you for your valuable feedback. We have updated our manuscript to include additional visual comparisons of the RestoreAgent. These enhancements provide a clearer demonstration of the effectiveness of our proposed method. Please refer to the updated **PDF file** we have uploaded, which contains these new visual results.

---

> > ### Comment · Reviewer_Ux72 · 2024-08-12
> >
> > Thank you to the authors for their detailed response. After reviewing all the explanations and the provided visual results, I can confirm that all of my concerns have been fully addressed.

---

### Official Review · Reviewer_NhBR · 2024-07-10

**Soundness:** 4
**Presentation:** 3
**Contribution:** 3
**Rating:** 6
**Confidence:** 5

**Summary:**

This paper presents an image restoration pipeline designed to handle various degradation types and levels by leveraging MLLM’s capabilities to select the appropriate model and determine the execution order. It begins with an analysis of why execution order and utilizing multiple models for different degradation levels are crucial for restoring complexly degraded images. The paper then constructs an instruction dataset and fine-tunes the MLLM. Experimental results demonstrate the effectiveness of the proposed restoration pipeline.

**Strengths:**

1.
This work presents a compelling analysis of complex image restoration. This insight is valuable given that degraded images in real-world scenarios often involve multiple types of degradation.
2.
This approach leverages the strengths of different models for handling specific noise levels, thereby eliminating the trade-off between generalization and performance.
3.
This paper formally defines the problem of handling multiple degradations and model selection in image restoration.
4.
Extensive experiments demonstrate superiority of such pipeline in processing degraded images with multiple degradations.

**Weaknesses:**

1.
In the introduction, it would be helpful to explain how the Multi-Level Learning Model (MLLM) excels at understanding different types and levels of image degradation. This will show why MLLM is well-suited for handling complex combinations of image degradation. Providing this clarity will make the benefits of using MLLM for image restoration more evident.
2.
When incorporating a new type of degradation, the cost extends beyond merely training the MLLM. Please also discuss the process of constructing training data for the newly added degradation and how it integrates with previously trained data.
3.
In lines 211-212, please clarify what the mean and standard deviation are calculated over. The subscript "i" is already used for degradation type and it might be clearer to use another character.

**Questions:**

1.
What if the degradation of the input image falls outside the predefined degradation scope? This could present a generalization issue, as the model might not perform well on unseen types or levels of degradation not covered in the predefined scope. Please discuss it.
2.
In Table 2, it would be clearer to highlight the best method for each evaluation criterion. Additionally, please specify which methods the ranking improvement is compared against for better context.

**Limitations:**

The authors have addressed the limitations.

---

> ### Author Rebuttal · Authors · 2024-08-04
>
> `Q1`: **In the introduction, it would be helpful to explain how the MLLM excels at understanding different types and levels of image degradation.**
>
> Thank you for your valuable suggestion. We will incorporate the following explanation into our introduction to clarify how MLLMs excel at understanding and handling different types and levels of image degradation.
>
> MLLMs are well-equipped to handle various types and levels of image degradation due to their extensive training on diverse datasets. This training enables them to develop a deep understanding of the relationships between different modalities, which is crucial for identifying and addressing complex degradation patterns.
> ﻿
> MLLMs excel in image restoration due to their robust reasoning abilities and advanced pattern recognition skills. They can analyze and interpret intricate details of various degradations, discerning subtle variations in noise, blur, compression artifacts, and other forms of image deterioration. This capability allows MLLMs to make precise decisions on applying the most appropriate restoration techniques, even when faced with complex combinations of degradation effects. For instance, when encountering an image with mixed noise, blur, and compression artifacts, an MLLM can accurately determine the optimal sequence and type of restoration techniques to apply.
> ﻿
> Furthermore, the adaptability of MLLMs allows them to quickly learn and optimize for new degradation scenarios through fine-tuning. This flexibility is essential for real-world applications where the types and combinations of degradations can vary widely. By continually updating their knowledge base, MLLMs can maintain high performance across a broad spectrum of image restoration tasks.
> ﻿
>
>
> `Q2`: **Please also discuss the process of constructing training data for the newly added degradation and how it integrates with previously trained data.**
>
> The process of constructing training data for new degradations follows the same methodology as for existing ones. For instance, if we were to add a new degradation type such as snow, we would randomly combine it with other existing degradations like noise and JPEG.
> Specifically, we experiment with different permutations and combinations of these degradations and find the best one to generate diverse training data.
> Once the new training data is generated, it is directly added to the original training dataset. This combined dataset is then used to fine-tune the model.
> Adding a new task requires a relatively small amount of data and has a quick training time. It allows for continuous improvement of the model without the need for complete retraining from scratch.
> We have updated our manuscript to include a more detailed discussion of this process.
>
>
>
> `Q3`: **In lines 211-212, please clarify what the mean and standard deviation are calculated over. The subscript "i" is already used for degradation type and it might be clearer to use another character.**
>
> We have revised lines 211-212. Our mean and standard deviation are calculated separately for each metric. Specifically, for each metric, we have results from all possible permutations and combinations restoration pipelines. We calculate the mean and variance using all these data points for each respective metric.
>
> As you correctly pointed out, using the subscript "i" for both degradation type and in the summation could lead to confusion. We have updated our notation to use different characters for clarity.
>
>
>
> `Q4`: **What if the degradation of the input image falls outside the predefined degradation scope?**
>
> We appreciate the reviewer's concern regarding the generalization capability of our model when faced with degradations outside the predefined scope. Here, we would like to further elaborate on this aspect.
>
> First of all, it is true that there might be generalization issues when encountering unseen types or levels of degradation. We acknowledge this limitation and have discussed it in detail in our manuscript.
>
> However, the primary and most significant contribution of our work is to demonstrate the feasibility and effectiveness of the proposed pipeline. The essence of our work lies in utilizing multimodal large models as agents for image restoration, which has shown promising results across various conditions. Our experiments show that the RestoreAgent performs exceptionally well in most complex scenarios, thus proving its viability.
>
> In addition, our approach allows for flexible integration of different tools. If the model encounters degradations beyond the predefined scope, additional tools can be seamlessly incorporated. This adaptability ensures that our pipeline remains robust even when faced with unexpected degradation types.
> Essentially, the generalization issue is more related to the specific image restoration models (tools) used within the pipeline, rather than the pipeline itself.
>
> Lastly, one of our future research directions is to significantly expand the scope of restoration tasks and incorporate a greater variety of restoration models. By doing so, we aim to enhance the system's capability to handle real-world complex degradations more effectively.
> This expansion will involve integrating models trained on a broader spectrum of degradation types and more generalizable restoration models, ensuring that the pipeline can generalize better to unseen conditions.
>
>
>
> `Q5`: **Specify Table 2**
>
> Thank you for your suggestion to improve the clarity of Table 2. We have updated Table 2 to highlight the best performing method for each evaluation criterion. We also have added a clear note to Table 2 specifying that the ranking improvement is compared against human experts.

---

> > ### Comment · Reviewer_NhBR · 2024-08-12
> >
> > Thanks the detailed response, my concerns were well addressed.

---

### Official Review · Reviewer_DJ9b · 2024-07-10

**Soundness:** 3
**Presentation:** 3
**Contribution:** 3
**Rating:** 8
**Confidence:** 4

**Summary:**

This paper introduces RestoreAgent, an innovative image restoration system that leverages multimodal large language models to autonomously handle images with multiple types of degradation. The system addresses limitations of existing all-in-one models and fixed task sequences by dynamically adapting to each image's specific degradations. RestoreAgent can identify degradation types, determine appropriate restoration tasks, optimize the execution sequence, select the most suitable models, and execute the restoration process autonomously. The authors present a method for constructing training data and demonstrate that RestoreAgent outperforms existing methods and human experts in handling complex image degradations.

**Strengths:**

1. This paper represents a innovation and a good contribution in image restoration and potentially opens up a new research direction for this area.
2. The motivation is strong. The authors effectively demonstrate the importance of task execution order and model selection in multi-task scenarios. The designed system adeptly addresses these issues.
3. Experimental results indicate that RestoreAgent's decision-making capabilities in handling complex degradations surpass those of human experts. This kind of pipeline also surpass all-in-one models.
4. The paper is generally well written and clear to understand.

**Weaknesses:**

1. The paper constructs a training dataset for training the multimodal large language model and a testing dataset as a benchmark for evaluating performance across multiple tasks. More details and explanations regarding the construction methods of these datasets would be beneficial.
2. Table 1 presents performance rankings using both ordinal and percentage forms. The definitions and explanations for these ranking forms are somewhat lacking, which might require readers to spend extra time understanding them. Clearer explanations would facilitate better comprehension.
3. The proposed Autonomous Restoration Agent represents a novel paradigm that is likely to encounter numerous new challenges. Beyond the issues already mentioned in the paper, the authors could consider discussing additional limitations and future research directions for this paradigm. This would help future researchers better follow and improve upon this work.

**Questions:**

1. The current method appears to predict all execution steps at once for a given input image. In Figure 3, each image has a dashed line pointing to the input. Does this imply that after each execution, the result can be fed back as input? (Based on my understanding, this system supports this) The paper seems to lack analysis and experiments related to this aspect. Could the authors provide more details on this part?
2. The authors have proposed a testing dataset to evaluate multi-task processing capabilities. Will this dataset be made publicly available to facilitate further research by other researchers?

**Limitations:**

See Weaknesses

---

> ### Author Rebuttal · Authors · 2024-08-04
>
> `Q1`. **More details regarding the construction methods of these datasets**
>
> Thank you for your feedback. We have significantly expanded the relevant sections in the revised version of our paper to offer a much more comprehensive explanation of our data preparation process.
>
> Regarding the training dataset, the following is part of the part of newly added detailed content:
>
> The training pairs construction begins with applying various types of degradation to an image. Subsequently, we determine the optimal restoration pipeline using model tools for processing. This involves generating all possible permutations of task execution sequences and model combinations, applying each pipeline to the degraded image, and assessing the quality of the restored outputs using a scoring function.  The model for different restoration tasks are shown in **Tbale 1**.
>
> **Table 1: Model tools for different restoration tasks.**
>
> | Task | Model Tools |
> |---|---|
> | Gaussian denosing | Restormer (trained on small / medium / large noise level) |
> | DeJPEG 	  | Restormer (trained on low / high quality factor)  |
> | Dehazing  | RIDCP |
> | Deraining | Restormer |
> | Motion deblurring | DeblurGANv2 |
> | Low-light enhancement | Retinexformer |
>
>  **Table 2** illustrates 5 scenarios incorporated into our dataset, designed to enhance the versatility and robustness of the RestoreAgent model:
>
> **Table 2: Five scenarios for dataset construction and corresponding examples.**
>
> | Index | Input | Answer | Function |
> |---|---|---|---|
> | 1 (Primary) | How to enhance the quality of this image? Execution history: None. | 1. Denoising, low noise level, 2. Dehazing, 3. DeJPEG, high quality factor. | Initiate full enhancement sequences for degraded images. |
> | 2 | How to enhance the quality of this image? Execution history: 1. Denoising, low noise level. | 1. Denoising, low noise level, 2. Dehazing, 3. DeJPEG, high quality factor | Dynamically adjust strategies based on intermediate results. |
> | 3 | How to enhance the quality of this image? Execution history: 1. Denoising, low noise level, 2. Dehazing | Rollback. | Identify and correct suboptimal steps through rollback mechanisms. |
> | 4 | How to enhance the quality of this image? Execution history: 1. Denoising, low noise level. Rollback from Dehazing. | 1. Denoising, low noise level, 2. DeJPEG, high quality factor, 3. Dehazing. | Avoid repetition of ineffective procedures post-rollback. |
> | 5 | How to enhance the quality of this image? Execution history: 1. Denoising, low noise level, 2. DeJPEG, high quality factor, 3. Dehazing. | Stop. | Recognize when image quality has reached its optimal state. |
>
> As for the testing dataset used as a benchmark, the details are presented in **Table 3**, which demonstrates our construction of various combinations of degradation types. Each image in the set contains a minimum of one and a maximum of four types of degradation, with the entire set comprising 200 images.
>
> **Table 3: Testset details.**
>
> | Degradation | # Images |
> |---|---|
> | Noise + JPEG | 50 |
> | Noise + Low light | 30 |
> | Motion Blur + Noise + JPEG | 30 |
> | Rain + Noise + JPEG | 20 |
> | Haze + Noise + JPEG | 30 |
> | Haze + Rain + Noise + JPEG | 20 |
> | Motion Blur + Rain + Noise + JPEG | 20 |
> | Total | 200 |
>
> ---
>
> `Q2`. **Explanations of "rankings" in Table 1**
>
> We've clarified the ranking system:
>
> For individual test sets:
> X/Y format: X is the average ranking, Y is total combinations.
> Example: 6.4/64 means average 6.4th best out of 64 options.
>
> For overall average:
> We use percentages due to varying combination counts of different restoration tasks.
> Process: Calculate percentage ranking per set, then average.
> Example: 12.9% means top 12.9% on average across datasets.
>
> ---
>
> `Q3`. **Additional limitations and future research directions**
>
> The primary limitation of our study is the confined scope of models and tasks examined. While our research offers valuable insights into RestoreAgent’s performance across several degradation scenarios, it does not encompass the full spectrum of restoration models or image degradation tasks currently available.
>
> Another limitation pertains to the limited generalization capability of current image restoration models. These models often exhibit a notable decrease in performance or fail to respond adequately when faced with even minor variations in image degradation patterns. This limitation greatly narrows our selection of model tools, requiring us to choose more robust and generalizable model tools. The challenge underscores a critical need in the field of image restoration: future models must go beyond simply overfitting training data.
>
> Our future work will focus on significantly expanding the range of image restoration models incorporated into our multimodal large language model. This expansion aims to enhance RestoreAgent’s capabilities across a broader scope of restoration tasks and degradation types. By integrating a more diverse set of state-of-the-art models, we seek to create a more comprehensive and versatile restoration framework.
>
> ---
>
> `Q4`. **Step-wise Re-planning and Rollback.**
>
> You are correct in your understanding. In the revised version of our paper, we have added a detailed discussion on this feature, which we call "Step-wise Re-planning and Rollback."
>
> In the answer to Q1, we have actually added some details (indexes 2, 3, and 4 of **Table 2**).
> **Table 4** shows experiments on a complex dataset with four image degradation types. While single prediction performs well, iterative step-wise replanning offers modest improvements. This indicates strong initial performance, with replanning serving as a refinement tool for incremental enhancements.
>
> **Table 4: Step-wise planning.**
>
> |            | balanced | ranking /64 |
> |---|---|---|
> | Human Expert | 5.42 | 21.2 |
> | RestoreAgent | 6.35 | 5.7  |
> | RestoreAgent + Step-wise | **6.38**| **4.5** |
>
> ---
>
> `Q5`. **Will this test set be made publicly available?**
>
> Yes, we will.

---

> > ### Comment · Reviewer_DJ9b · 2024-08-09
> >
> > Thanks to the authors for their efforts and replies. The authors' rebuttal addressed my questions about implement details. It's refreshing to tackle image restoration with the Agent paradigm, and I'm sure this work can inspire the community a lot! Overall, this work is quite good and I wish to see the follow up work.

---

### Author Rebuttal · Authors · 2024-08-06

Dear AC and all reviewers,

We sincerely appreciate your time and efforts in reviewing our paper. We are glad to find that reviewers recognized the following merits of our work:

- **Innovative contribution and strong motivation [DJ9b, NhBR, Ux72, 4Dbc]**:
The proposed RestoreAgent addresses the challenges of image restoration by effectively demonstrating the importance of task execution order and model selection in multi-task scenarios. This novel approach opens up new research directions in the field.

- **Impressive performance [DJ9b, NhBR, Ux72, 4Dbc]**: Experimental results indicate that RestoreAgent's decision-making capabilities in handling complex degradations surpass those of human experts and existing all-in-one models.


- **Well-written and clear [DJ9b, Ux72, 4Dbc]**: The paper is generally well written and clear to understand, with comprehensive analysis and visualization.



We also thank all reviewers for their insightful and constructive suggestions, which help further improve our paper. In addition to the pointwise responses below, we summarize the major revision in the rebuttal according to the reviewers’ suggestions:


- **Detailed data construction process [DJ9b, 4Dbc]**:
We have added detailed explanations on the construction methods for the training and testing datasets.


- **Discussion on Limitations and Future Directions [DJ9b, NhBR, Ux72]**:
We have expanded the discussion on the potential challenges and future research directions, including the integration of new degradation types, generalization to unseen degradations, and the limitations of existing frameworks.


- **Manuscript update [DJ9b, Ux72, 4Dbc, NhBR]**: We have included more visual comparisons, inference time, experimental results, and discussions in the main paper and Appendix, Furthermore, we have clarified descriptions that were previously unclear.


We hope our pointwise responses below can clarify all reviewers' confusion and address the raised concerns. We thank all reviewers' efforts and time again.

Best,
Authors

---

### Decision · Program_Chairs · 2024-09-25

**Decision:**

Accept (poster)

**Comment:**

This paper explores the multimodal large language models for image restoration. All reviewers are in favor of this manuscript. The AC agrees with the recommendations of reviewers.